# Attention Head Purification: A New Perspective to Harness CLIP for Domain Generalization

## Abstract

Domain Generalization (DG) aims to learn a model from multiple source domains to achieve satisfactory performance on unseen target domains. Recent works introduce CLIP to DG tasks due to its superior image-text alignment and zeros-shot performance. Previous methods either utilize full fine-tuning or prompt-learning paradigms to harness CLIP for DG tasks. Those works focus on avoiding catastrophic forgetting of the original knowledge encoded in CLIP but ignore that the knowledge encoded in CLIP in nature may contain domain-specific cues that constrain its domain generalization performance. In this paper, we propose a new perspective to harness CLIP for DG, *i.e.,* attention head purification. We observe that different attention heads may encode different properties of an image and selecting heads appropriately may yield remarkable performance improvement across domains. Based on such observations, we purify the attention heads of CLIP from two levels, including *task-level purification* and *domain-level purification*. For task-level purification, we design head-aware LoRA to make each head more adapted to the task we considered. For domain-level purification, we perform head selection via a simple gating strategy. We utilize MMD loss to encourage masked head features to be more domain-invariant to emphasize more generalizable properties/heads. During training, we jointly perform task-level purification and domain-level purification. We conduct experiments on various representative DG benchmarks. Though simple, extensive experiments demonstrate that our method performs favorably against previous state-of-the-arts.

## 1 Introduction

Deep learning has attained remarkable success on various downstream tasks in computer vision, typically under the assumption that both training and test samples are identically distributed. However, in practice, test data distributions (target) are usually different from the training ones (source). In such cases, the performance of deep neural networks may degenerate severely on target, showing a poor domain generalization ability. To mitigate the domain shift, a series of Domain Generalization (DG) methods (Carlucci et al., 2019; Cha et al., 2022; Gulrajani & Lopez-Paz, 2020; Kim et al., 2022; Li et al., 2018a;b; Zhou et al., 2021) are proposed to transfer the knowledge learned from multiple source domains to unseen target domains, via domain-invariant learning (Wang et al., 2022; Jia et al., 2020; Li et al., 2020; 2018c; Shao et al., 2019; Wang et al., 2021), meta-learning techniques (Sankaranarayanan & Balaji, 2023; Balaji et al., 2018b;a; Li et al., 2018a), or specifically-designed data augmentations (Volpi et al., 2018; Qiao et al., 2020; Zhou et al., 2020a). The remarkable progress achieved by those works can be largely attributed to a well-initialized feature extractor provided by ImageNet-pretrained (Huh et al., 2016) backbones. Recent proposed Contrastive Language–Image Pre-training (*i.e.,* CLIP (Radford et al., 2021)) learns from large amounts of image-text pairs (Jia et al., 2021) and demonstrates impressive zero-shot learning performance. It may potentially serve as a better foundation for mitigating the performance gap across domains for specific downstream tasks.

Despite superior zero-shot classification performance of CLIP, it is not trivial to harness CLIP for specific domain generalization tasks beyond its zero-shot classification ability. Directly fine-tuning CLIP may obtain even worse performance than zero-shot classification (Pham et al., 2023; Wortsman et al., 2022). Previous works tackle this issue mainly by introducing various regularizations to avoid

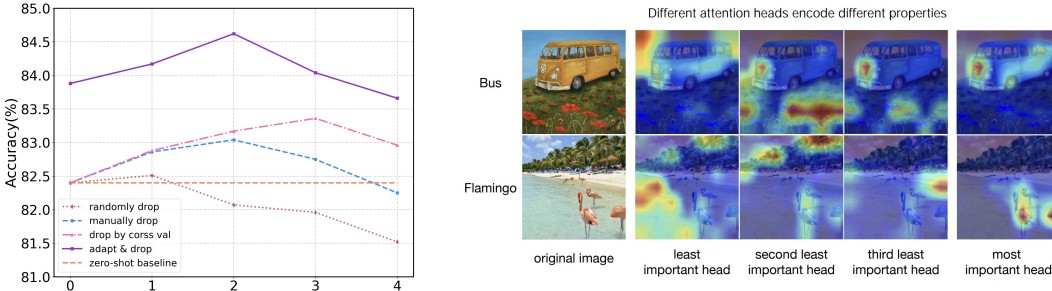

Figure 1: **Left**: We use different strategies to evaluate the importance of each attention head on domain generalization and drop the least important ones to see how accuracy changes. The strategies we adopt include "randomly drop(–◇–)", "manually drop(–●–)", "drop by cross-validation(–▲–)", and "adapt & drop(–■–)". Details of strategies can be found in Appendix. Via appropriately dropping heads, CLIP's domain generalization performance can be improved. **Right:** Attention map (Chefer et al., 2021) generated by specific heads. The middle columns are from least important heads determined by the cross-validation strategy. They all capture lots of background information. The last column represents the most important one whose attention map mainly focuses on the object itself. The experiments are conducted on OfficeHome (Venkateswara et al., 2017). Best viewed in color.

the forgetting of original knowledge encoded in CLIP. For example, Shu et al. (2023) fully fine-tunes CLIP's image encoder with modified contrastive loss to mitigate overfitting and proposes beta moving average to perform a temporal ensembling along the fine-tuning trajectory. Cha et al. (2022) proposes mutual information regularization between the original CLIP and the fine-tuned one to prevent the two models from deviating too much. Although those works achieve improved domain generalization performance, a natural question arises: *is the way to avoid knowledge forgetting sufficient to harness CLIP for domain generalization?*

Recent work (Gandelsman et al., 2023) shows that different attention heads of CLIP's image encoder (ViT-based) may encode different properties of an image. Inspired by this work, we conduct ablation experiments to verify the effectiveness of each head in the context of domain generalization. Specifically, we utilize different ways to evaluate the importance of attention heads on domain generalization (*e.g.,* manual evaluation, evaluation by cross-validation, *etc.*) and drop the least important ones. As shown in Figure 1(a), we observe that via appropriately dropping some attention heads, we may achieve a much better domain generalization performance. This phenomenon implies that not all attention heads are domain generalizable and some heads may harm the domain generalization ability of CLIP, which means those heads may contain non-generalizable cues (*i.e.,* domain-specific cues). To further illustrate our findings, we provide attention visualizations in Figure 1(b). We observe that the attention maps (Chefer et al., 2021) of the least important heads mainly highlight task-irrelevant regions like background, while the most important head pays more attention to the object itself. As a result, simply avoiding the knowledge forgetting of CLIP may not be optimal for domain generalization tasks. We need to purify the attention heads to make CLIP more task-adapted and domain-generalizable.

In this paper, we propose a new perspective to harness CLIP for domain generalization tasks, *i.e.,* through attention head purification. Specifically, we perform two kinds of attention head purification during training, which are named *task-level purification* and *domain-level purification*. For task-level purification, we aim to purify the attention head of CLIP to maintain more task-related knowledge. Technically, we adopt LoRA to realize this goal. Different from conventional LoRA implementation, we design head-aware LoRA (HA-LoRA) to purify and adapt each head more accordingly. For domain-level purification, we aim to purify the attention heads of CLIP to make the resulting features more invariant across domains. We design a simple learnable gating strategy to select the heads that most benefit the domain generalization performance. To realize this, in addition to the cross-entropy loss on source images, we utilize Maximize Mean Discrepancy (MMD) loss to encourage the gates to emphasize more domain-invariant head features. Note that we do not utilize the gradients of MMD loss to update the HA-LoRA. In this way, we may decouple task-level and domain-level purification to an extent, *i.e.,* we expect HA-LoRA to focus only on encoding rich *task-related* properties, while leaving the goal of selecting domain-invariant properties to domain-level purification. During training, we jointly train the head-aware LoRA and the gates of attention heads to perform the task

and domain-level purification simultaneously. Pre-trained parameters in the image encoder and the text encoder of CLIP are frozen throughout the training. We conduct extensive experiments on various representative domain generalization benchmarks including Office-Home (Venkateswara et al., 2017), DomainNet (Peng et al., 2019), PACS (Li et al., 2017), VLCS (Torralba & Efros, 2011) and TerraIncognita (Beery et al., 2018). All those experiments verify the superiority of our proposed method.

In a nutshell, our contributions can be summarized as **1)** We observe that not all attention heads of CLIP are domain generalizable in terms of specific tasks and propose a new perspective to harness CLIP for DG through attention head purification, *i.e.,* optimizing attention heads to make them more task-adapted and domain generalizable. **2)** We decouple the attention head purification from two levels including task-level purification and domain-level purification. We design head-aware LoRA and learnable gating strategy to perform the two kinds of purification respectively. **3)** Extensive experiments on representative domain generalization benchmarks demonstrate that our method performs favourably against the previous state-of-the-art.

## 2 RELATED WORKS

**Vision-Language Pre-training.** Vision-Language models(VLMs) connect images and texts through a common embedding space to enable cross-modal learning (Frome et al., 2013; Socher et al., 2013; Elhoseiny et al., 2013). Recent advances employ architectures with better representation learning abilities such as Transformer (Vaswani et al., 2017) and webscale training datasets and build stronger vision-language pre-trained models. One type of approach learns the common embedding space by masked language modeling or masked region prediction (Tan & Bansal, 2019; Su et al., 2019; Kim et al., 2021). Another typical type of vision-language pretraining is contrastive image-language pre-training, such as CLIP (Radford et al., 2021) and ALIGN (Jia et al., 2021). Recent research also seeks to improve the pre-training paradigm, such as using additional supervision (Li et al., 2021; Mu et al., 2022), employing pre-trained image encoders (Zhai et al., 2022), and adding cross-modal and in-modal consistency constraints (Goel et al., 2022). In this paper, instead of designing better pretraining techniques, we aim at utilizing recent advances in vision-language pre-trained models such as CLIP and achieving better generalization performance.

**Domain Generalization(DG).** Domain generalization aims to learn generalized representations from multiple source domains that can generalize well on arbitrary unseen target domains. Most DG methods perform domain alignment (Ganin & Lempitsky, 2015; Jia et al., 2020; Li et al., 2018c; Shao et al., 2019; Li et al., 2020; Wang et al., 2021; Ouyang & Key, 2021), to learn domain invariant features by reducing the distance of distributions across multiple domains. Specifically, to achieve domain-invariance, Jia et al. (2020); Li et al. (2018c); Shao et al. (2019); Ouyang & Key (2021); Ganin & Lempitsky (2015) imply adversarial learning, Ouyang & Key (2021) minimizes maximum mean discrepancy (MMD), and Sankaranarayanan & Balaji (2023); Balaji et al. (2018a); Dou et al. (2019) utilize meta-learning techniques. In addition, Zhou et al. (2020b); Cubuk et al. (2020); Li et al. (2022); Kang et al. (2022); Zhou et al. (2021); Nuriel et al. (2021); Li et al. (2023); Guo et al. (2023) use domain augmentation to enrich the style diversity of source data at image level or feature level. The remarkable progress achieved by those works can be largely attributed to a well-initialized feature extractor provided by ImageNet pre-training (Huh et al., 2016). Recent proposed Vision-language pre-trained models such as CLIP exhibit impressive zero-shot generalization ability. Its zero-shot performance exceeds the aforementioned methods in various DG tasks. In this paper, we focus on leveraging CLIP to further improve domain generalization performance.

**CLIP-based Domain Generalization.** Despite the good zero-shot performance, research finds that directly finetuning CLIP models with task-specific data will damage the alignment in joint vision-language spaces (Zhang et al., 2023b) and harm CLIP's domain generalization ability (Radford et al., 2021; Wortsman et al., 2022). Recent advances explore adapting CLIP to downstream tasks without affecting its generalization ability by adapter learning (Gao et al., 2024; Zhang et al., 2021), model ensemble (Wortsman et al., 2022), regularized fine-tuning (Wortsman et al., 2022; Cha et al., 2022; Shu et al., 2023; Lew et al., 2023), distillation (Hémadou et al.; Huang et al., 2023), and prompt learning (Cho et al., 2023; Zhou et al., 2022b; Zhang et al., 2023a; Niu et al., 2022; Bose et al., 2024; Cheng et al., 2024; Bai et al., 2024). Different from most existing works that use regularization to avoid knowledge forgetting of CLIP, we propose attention head purification to harness CLIP for DG.

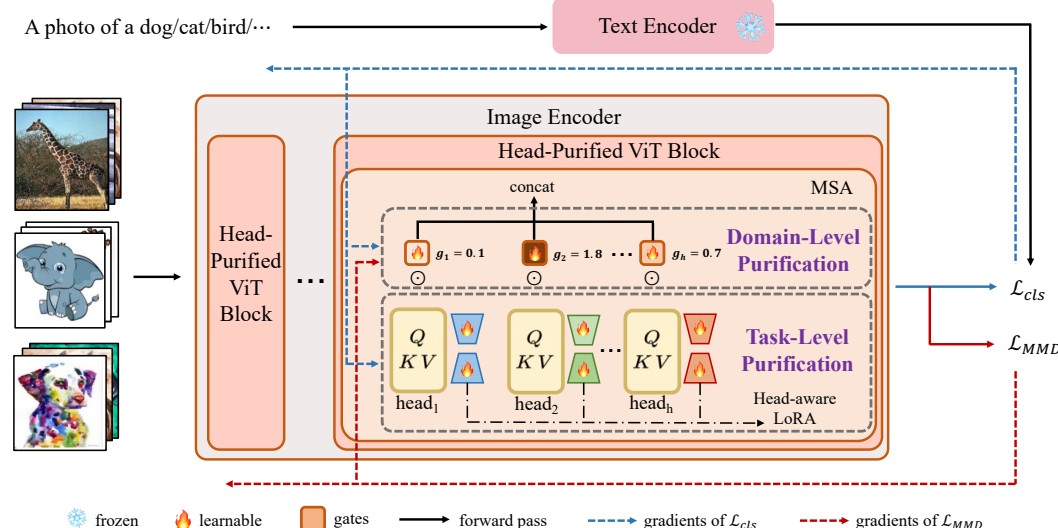

Figure 2: The architecture of Attention Head Purification. We design head-aware LoRA to perform task-level purification and domain-invariant gating strategy to perform domain-level purification. Further, we minimize $\mathcal{L}_{cls}$ (Section 3.2.1) to update head-aware LoRA and minimize $\mathcal{L}_{cls}, \mathcal{L}_{\mathrm{MMD}}$ (Section 3.2.2) to update the gates of heads.

## 3 METHOD

### 3.1 PRELIMINARY

**Background.** We aim to tackle the domain generalization problem based on CLIP. Specifically, suppose we have multiple source domains for training. Our goal is to adapt CLIP with labeled samples from source domains to make it perform well on unseen target domains. Usually, the distribution of target domain data is distinct from that of each source domain.

**CLIP Revisiting.** CLIP is a large-scale visual-language model that consists of an image encoder and a text encoder. CLIP is trained with 400 million web-scraped image-text pairs (Radford et al., 2021). The contrastive loss is imposed to encourage the features from paired image-text to be more aligned than unpaired ones. As a result, CLIP is readily adopted to downstream tasks even without any fine-tuning, showing impressive zero-shot classification ability. Specifically, given $C$ classes with their class names, we may construct a text description for each class, *e.g.,* "A photo of a #classname". For a specific image, we extract its image feature, and compare the extracted image feature with text features of different classes. Then, the class of text feature which has a highest cosine similarity with the image feature is viewed as the predicted label of the image. Formally, the zero-shot classification process can be represented as $\hat{y} = \mathrm{argmax}_c \cos(I, T_c)$, where $I$ denote the image feature, $T_c$ denotes the text feature of class $c \in \{0, 1, 2, \cdots, C-1\}$, and $\hat{y}$ is the predicted label for the image.

**LoRA Revisiting.** Low-rank adaptation (LoRA) (Hu et al., 2021) is one of the most popular parameter-efficient fine-tuning (PEFT) methods. It assumes that the changes of parameters lie in a low-rank space when the model is fine-tuned on a downstream task. Specifically, for a linear layer with the input dimension $d_I$ and the output dimension $d_O$, we represent its weight with $W^{d_O \times d_I}$. Then LoRA reparametrizes the pre-trained weight $W$ by expanding a branch with two matrices, $A \in R^{d_O \times r}$ and $B \in R^{r \times d_I}$. Typically, $r$ is much smaller than the input dimension $d_I$ and output dimension $d_O$, making $A$ a dimension increasing matrix and $B$ a dimension reduction matrix. Finally, LoRA modifies the forward propagation in this linear layer as $o = We + ABe$ where $e$ and $o$ denote the input and output features of this layer respectively. During adaptation to the downstream tasks, we freeze the pre-trained weight $W$ and only update $A$ and $B$.

### 3.2 ATTENTION HEAD PURIFICATION OF CLIP

**Overall Framework.** The image encoder of CLIP encodes rich properties of an image into image features. For a specific downstream task, not all properties are beneficial, *i.e.,* some properties may

not be task-related, and some properties may not be domain-invariant. As shown in Gandelsman et al. (2023), different attention heads may encode different properties of an image. From this point of view, we propose to purify the attention heads of CLIP from two levels, including task-level purification (detailed in Section 3.2.1) and domain-level purification(detailed in Section 3.2.2). For task-level purification, we purify the attention head of CLIP using our designed head-aware LoRA (HA-LoRA) to focus on task-related properties. For domain-level purification, we design a simple learnable gating strategy to emphasize generalizable heads while restraining domain-sensitive heads. During training, we add HA-LoRA and head gating into certain layers of CLIP's image encoder as shown in Figure 2 to conduct task-level purification and domain-level purification simultaneously.

### 3.2.1 TASK-LEVEL PURIFICATION WITH HEAD-AWARE LoRA

From the task-level, we use LoRA to purify each attention head, *i.e.,* encouraging the heads to focus on task-related patterns while ignoring task-irrelevant ones. Technically, for a linear projection $W$, we may add a branch which sequentially multiplies a dimension reduction matrix $B \in \mathbb{R}^{r \times d_I}$ and a dimension increasing matrix $A \in \mathbb{R}^{d_O \times r}$. Then, the forward propagation of this linear layer is modified as

$$o = We + ABe = W'e, \tag{1}$$

where $W' = W + AB$ denotes the adapted weight, $e$ denotes the input of the projection, and $o$ denotes the output. We only use LoRA to adapt the linear projections of query ($Q$) and value ($V$) in multi-head self-attention (MSA) blocks. To improve the effect of task-level purification and facilitate the following domain-level purification, we propose using head-aware LoRA instead of the conventional LoRA in our framework.

**Head-aware LoRA.** For the pre-trained weight $W \in \mathbb{R}^{d_O \times d_I}$ of the $Q$ or $V$ projection in a MSA block, we can split $d_O$ into $H$ groups (*i.e.,* $W_1, W_2, \cdots, W_H$) where $H$ denotes the number of attention heads in a MSA block. As a result, $W_h \in \mathbb{R}^{n \times d_I}, h \in \{1, 2, \cdots, H\}$ where $n = d_O/H$. We split the matrix $A$ in the same way and obtain $A_1, A_2, \cdots, A_H$ where $A_h \in \mathbb{R}^{n \times r}$. Then the adapted weight $W'$ of the conventional LoRA can be reformulated as

$$W' = W + AB = W + \begin{pmatrix} A_1 B \\ A_2 B \\ \vdots \\ A_H B \end{pmatrix} = \begin{pmatrix} W_1 + A_1 B \\ W_2 + A_2 B \\ \vdots \\ W_H + A_H B \end{pmatrix}. \tag{2}$$

From Eq. (2), we observe that in the conventional LoRA, different from $A_h$ which is distinct with respect to different heads, $B$ is shared by all the heads. As a result, purifying one head may interfere with the other head, rendering the head purification less effective.

To mitigate such interference between different heads, we propose head-aware LoRA, denoted as HA-LoRA. As shown in Task-Level Purification module in Figure 2, we set independent $B_h \in \mathbb{R}^{r \times d_I}$ for each head. The adapted weight $W'$ for HA-LoRA can be represented as:

$$W' = W + \begin{pmatrix} A_1 B_1 \\ A_2 B_2 \\ \vdots \\ A_H B_H \end{pmatrix} = \begin{pmatrix} W_1 + A_1 B_1 \\ W_2 + A_2 B_2 \\ \vdots \\ W_H + A_H B_H \end{pmatrix} \tag{3}$$

Different from Tian et al. (2024), which also uses independent $B$ to handle different tasks, we use independent $B$ to modulate different heads. The two approaches are technically similar but differ in their underlying motivations.

We follow the vision-language contrastive learning strategy to update the parameters of HA-LoRA. For each class $c$, we manually generate a text describing it, *e.g.,* "A photo of a #classname". We then get the text embedding of each class $T_c$ extracted by the text encoder. Then, we impose a contrastive loss between image features and the text features of different classes to encourage the image feature to be more aligned with the text feature of ground-truth label, *i.e.,*

$$\mathcal{L}_{cls} = -\frac{1}{N} \sum_{i=1}^{N} \log \frac{\exp(\cos(s_i, T_y)/\tau)}{\sum_{c=1}^{C} \exp(\cos(s_i, T_c)/\tau)} \tag{4}$$

where $s_i$ denotes the image feature for the $i$-th sample, $T_c$ denotes the text feature of class $c$, $y$ is the ground-truth label for the $i$-th image and $\tau$ is the temperature hyper-parameter.

### 3.2.2 DOMAIN-LEVEL PURIFICATION WITH DOMAIN-INVARIANT GATING

By task-level purification, we can wipe off task-unrelated properties, making each head more adapted to current task. After this, we also need to perform domain-level purification, aiming to maintain or emphasize the most generalizable or invariant attention heads across domains. To realize this, we design a domain-invariant gating (DIG) scheme. Specifically, in a MSA block, we set a series of learnable gates $g_1, g_2, \cdots, g_H$, the number of which equals the number of attention heads $H$. To evaluate the relative importance of each head, we apply softmax operation to the gates, *i.e.,* $\hat{g}_1, \hat{g}_2, \cdots, \hat{g}_H = \text{Softmax}(g_1, g_2, \cdots, g_H)$. Then, we apply the gates to the features of different heads which are the outputs of the scaled dot-product attention operation. We concatenate the gated features from all the heads and obtain $f^g$ as

$$f^g = \gamma[\hat{g}_1 f_1, \hat{g}_2 f_2, \cdots, \hat{g}_H f_H] \tag{5}$$

where $f_1, f_2, \cdots, f_H$ denote the features of different attention heads. The $\gamma$ equals to the number of heads, which is used to compensate the scale changes after softmax operation.

During training, in addition to $\mathcal{L}_{cls}$, we also utilize Maximum Mean Discrepancy (MMD) loss (Long et al., 2015; 2017) to measure the distribution discrepancy of features between different source domains, *i.e.,*

$$\text{MMD}(S^p, S^q) = \frac{1}{N^2} \sum_{i=1}^{N} \sum_{i'=1}^{N} k(s_i^p, s_{i'}^p) + \frac{1}{M^2} \sum_{j=1}^{M} \sum_{j'=1}^{M} k(s_j^q, s_{j'}^q) - \frac{2}{MN} \sum_{i=1}^{N} \sum_{j=1}^{M} k(s_i^p, s_j^q)$$

$$\mathcal{L}_{\text{MMD}} = \frac{2}{d(d-1)} \sum_{p=1}^{d-1} \sum_{q=p+1}^{d} \text{MMD}(S^p, S^q) \tag{6}$$

where $S^p = \{s_i^p\}_{i=1}^{N}$ are image features of $p$-th source domain output by CLIP's image encoder, $N$ denotes the number of samples for domain $p$ within a mini-batch, $d$ denotes the number of source domains and $k$ denotes the Gaussian kernel (Long et al., 2015; 2017).

In this way, if the gates select/emphasize heads which are more generalizable, the resulting features are more generalizable, rendering $\mathcal{L}_{\text{MMD}}$ small, otherwise the $\mathcal{L}_{\text{MMD}}$ will be large. Thus, minimizing $\mathcal{L}_{\text{MMD}}$ will encourage the gates to update towards selecting/emphasizing the most generalizable or domain-invariant heads.

### 3.3 OBJECTIVE

Since both task-level purification and domain-level purification contribute to CLIP's domain generalization performance, we want to combine them to achieve further performance improvement. We perform end-to-end training that simultaneously conducts task-level purification and domain-level purification as shown in Figure 2. The final optimization objective is as follows

$$\min_{\theta_1, \theta_2} \mathcal{L}_{cls} + \min_{\theta_2} \alpha \mathcal{L}_{\text{MMD}}, \tag{7}$$

where $\theta_1 = \{A_1, A_2, \cdots, A_H, B_1, B_2, \cdots, B_H\}$, and $\theta_2 = \{g_1, g_2, \cdots, g_H\}$. The $\alpha$ controls the strength of MMD loss.

As shown in Eq. (7), we do not use MMD loss to update HA-LoRA. In this way, we may decouple task-level and domain-level purification to an extent, *i.e.,* we expect HA-LoRA to focus only on encoding rich *task-related* properties, while leaving the goal of selecting domain-invariant properties to domain-level purification. For inference, we can merge HA-LoRA with the original weights of CLIP, eliminating extra memory overhead.

**Head-aware LoRA *vs.* LoRA.** Note that when jointly performing task-level and domain-level purification, head-ware LoRA is more beneficial to the domain-level purification than the convention LoRA. It is because different to conventional LoRA, head-aware LoRA owns different learnable parameters across different heads, which avoids the interference between different heads. Through such a head-interference elimination, the DIG may more independently and accordingly emphasize or restrain specific heads (see Section 5.1 for empirical details).

### 3.4 COMBINE WITH PROMPT-LEARNING METHODS

For classification with CLIP, we follow the general practice that comparing the similarity scores between the image feature and text features of difference classes. To generate the text feature of a

specific class, we need to conduct a text description which contains a prompt and the name of a class, and forward it through the text encoder of CLIP. The quality of prompt also affects the generalization ability of CLIP. Thus, many previous works, *e.g.,* CoOp (Zhou et al., 2022b), CoCoOp (Zhou et al., 2022a) and DPL (Zhang et al., 2023a) try to optimize the prompt to improve the generalization ability of CLIP. In this paper, we focus on how to extract more generalizable image features from CLIP, which is orthogonal to the previous prompt-learning methods. Thus, in practice, we may combine our method with previous prompt-learning method to maximize the generalization ability of CLIP across different domains, *i.e.,* we may utilize previous prompt-learning methods to optimize the prompt and employ our attention head purification to optimize the image features.

## 4 DATASETS AND IMPLEMENTATION DETAILS

### 4.1 DATASETS

For comparison, we evaluate our method on five representative datasets, including Office-Home (Venkateswara et al., 2017), PACS (Li et al., 2017), VLCS (Torralba & Efros, 2011), TerraIncognita (Beery et al., 2018) and DomainNet (Peng et al., 2019). **OfficeHome** (OH) includes 4 domains with 15,588 examples from 65 classes. **VLCS** includes 4 domains with 10,729 examples from 5 classes. **PACS** includes 4 domains with 9,991 examples from 7 classes. **DomainNet** (DN) includes 6 domains with 586,575 examples from 345 classes. **TerraIncognita** (TI) includes 4 domains with 24788 examples from 10 classes. Following Xu et al. (2021); Ganin & Lempitsky (2015), we adopt the typical leave-one-domain-out protocol for evaluation, *i.e.,* each time we select one out of available domains as the target for testing and the remaining domains as sources for training. The average accuracy across all target choices is reported.

### 4.2 IMPLEMENTATION DETAILS

We use the CLIP pre-trained model with ViT-B/16 as the image encoder. We only tune the image encoder. The text encoder of CLIP is kept frozen throughout the training. The batch size is set to 36. We use the AdamW as the optimizer with the cosine learning rate strategy for all datasets. We use a learning rate of $5 \times 10^{-5}$ for updating head-aware LoRA and $1 \times 10^{-3}$ for optimizing the head gates. For each run, we train the model for 40 epochs. We report the average result over three runs with different random seeds. We set the temperature $\tau$ to 0.01 which is the same as the pre-trained model. The $\alpha$ is set to 0.2 and kept the same across all the datasets. We purify the attention heads in all layers of the image encoder, and impose the MMD loss on each layer. For all the experiments, the rank of our head-aware LoRA is set to 8 for the last two layers and 2 for the rest layers.

## 5 EXPERIMENTAL RESULTS

We evaluate the proposed method for domain generalization classification task. We first conduct an extensive ablation study to validate key components of our framework. Second, we show that the proposed method can be combined with prompt-learning techniques and obtain significant performance gain. This demonstrates that our attention head purification technique is complementary to a wide range of prompt-learning strategies and provides additional benefits. Third, we show that our model performs favorably against previous state-of-the-art approaches. Finally, we visualize the attention maps of the overall model and specific heads.

### 5.1 ABLATION STUDY

**Task-level purification and domain-level purification cooperates to improve the generalization ability of CLIP.** In Table 1 (Left), we verify the effectiveness of task-level purification with head-aware LoRA ("HA-LoRA") and domain-level purification with domain-invariant gating ("DIG"). We observe that both task-level purification and domain-level purification contribute to the performance improvement compared to the zero-shot baseline (the first line in Table 1 (Left)). When combining both of them, we may further improve the domain generalization performance, obtaining 4.6% gain on OfficeHome and 3.4% gain on DomainNet compared to the zero-shot baseline. All those results verify the effectiveness of our proposed task-level purification and domain-level purification operations.

**Head-aware LoRA eliminates interference between different heads, benefiting subsequent head selection in domain-level purification.** In Table 1 (Right), we investigate the effect of our proposed HA-LoRA compared with the conventional LoRA. We find that without performing domain-level purification (*i.e.,* without using DIG), the result of HA-LoRA is only slightly better than that of the

Table 1: **Left:** Effect of task/domain-level purification. **Right:** Head-aware LoRA *vs.* original LoRA.

| HA-LoRA | DIG | OfficeHome | DomainNet | | LoRA | HA-LoRA | DIG | OfficeHome | DomainNet |
|---|---|---|---|---|---|---|---|---|---|
| | | 82.4 | 57.7 | | ✓ | | w/o. | 84.8 | 58.7 |
| ✓ | | 85.0 | 58.8 | | | ✓ | w/o. | 85.0(+0.2%) | 58.8(+0.1%) |
| | ✓ | 83.6 | 58.2 | | ✓ | | w. | 86.0 | 59.7 |
| ✓ | ✓ | 87.0 | 61.1 | | | ✓ | w. | 87.0(+1.0%) | 61.1(+1.4%) |

conventional LoRA (around 0.1% gain). However, when jointly optimizing domain-invariant gates and LoRA/HA-loRA, HA-LoRA achieves a remarkable improvement compared to the conventional LoRA (more than 1% gain). This is because the proposed head-aware LoRA can effectively eliminate the interference between different heads. Through interference elimination, the DIG may more independently and accordingly emphasize or restrain specific heads, rendering the domain-level purification more effective.

**Decoupling task-level purification and domain-level purification is beneficial.** As discussed in Section 3.2.2, we adopt MMD loss to encourage the head gates in DIG to update towards making the features invariant across domains. Technically, we may also adopt MMD loss to update HA-LoRA to encourage HA-LoRA to encode both task-related and domain-invariant properties. But we find this will harm the generalization performance. In Table 2, we compare the training without imposing any MMD loss (the first line), using MMD loss to update HA-LoRA (the second line), using MMD loss to update both the head gates and HA-LoRA (the third line) and our solution that uses MMD loss to update the head gates only (the last line). We observe that our solution obviously outperforms the one without any MMD loss, showing that MMD loss contributes to selecting/emphasizing the most domain-generalizable attention heads. When applying MMD loss to update HA-LoRA, the performance decreases. It is because applying MMD to HA-LoRA may encourage HA-LoRA to be both task-adapted and domain-invariant. Such a coupled optimization is more difficult.

**Joint purification training is the first choice.** In Table 3 (Left), we investigate the effect of different training strategies, including ours which jointly trains HA-LoRA and DIG, alternatively training DIG and HA-LoRA (denoted as alternative), training DIG first and then training HA-LoRA with fixed DIG (domain→task), and training HA-LoRA first and then training DIG with fixed HA-LoRA (task→domain). We observe that ours achieves the best results among different training strategies.

Table 2: Performance of using $\mathcal{L}_{\mathrm{MMD}}$ to update different modules. Results show that we may not obtain domain generalizable features directly through encouraging domain-invariant HA-LoRA.

| Modules that updated by $\mathcal{L}_{\mathrm{MMD}}$ | OfficeHome | DomainNet | PACS |
|---|---|---|---|
| Neither | 86.2 | 60.5 | 96.9 |
| HA-LoRA | 85.8 | 58.6 | 96.1 |
| HA-LoRA + DIG | 86.0 | 59.9 | 96.7 |
| DIG | 87.0 | 61.1 | 98.1 |

Table 3: **Left:** Effect of different training strategies. **Right:** Sensitivity to the ratio $\alpha$ of MMD loss term. The experiments are conducted on OfficeHome. The trends are similar for the other datasets.

| Method | OfficeHome | | Values of $\alpha$ | OfficeHome |
|---|---|---|---|---|
| jointly | 87.0 | | 0.0 | 86.2 |
| alternative | 86.7 | | 0.1 | 86.7 |
| two-stage(Task → Domain) | 85.8 | | 0.2 | 87.0 |
| two-stage(Domain → Task) | 85.1 | | 0.3 | 86.9 |
| | | | 0.5 | 86.7 |

**Sensitivity to the ratio $\alpha$ of MMD loss term.** In Table 3 (Right), we evaluate how the performance changes as $\alpha$ increases. We observe as $\alpha$ increases, the accuracy firstly increases and then decreases, exhibiting a typical bell curve. This phenomenon shows the regularization effect of MMD loss term. Besides, within a vast range of $\alpha$, the accuracy only slightly fluctuates. Note that the trends are similar across all the datasets. The results show that our method is relatively robust to the choices of $\alpha$.

## 5.2 IMPROVEMENT BEYOND PROMPT-LEARNING METHODS

In Table 4, we combine our method with representative prompt-learning DG methods. Besides, we also report numbers obtained by our method with manually designed prompt "A photo of a"

(same as zero-shot CLIP baseline). We observe that even without any prompt optimization, our method achieves remarkable improvement compared to zero-shot CLIP baseline (around 8% gain). Previous prompt-learning DG methods can be roughly categorized into two groups. One group, *e.g.,* DUPRG and PromptStyler, only utilizes text features to optimize the prompt. The other group, *e.g.,* CoOp, CoCoOp, DPL and STYLIP, utilizes the alignment between image features and text features to optimize the learnable prompt. Thus, for different groups, we combine attention head purification with prompt-learning in different ways. For the first group, we sequentially perform prompt optimization and attention head purification, *i.e.,*, we firstly obtain optimized prompt with prompt-learning method and then utilize the optimized prompt in attention head purification learning. For the second group, we jointly learn attention head purification and optimize the prompt. As shown in Table 4, combining with attention head purification can consistently improve the generalization performance of CLIP beyond the prompt-learning methods, *e.g.,* combining attention head purification with PromptStyler yields the best result (79.0% average accuracy) with 5.5% improvement beyond PromptStyler.

Table 4: Improvement beyond prompt-learning methods. [†] indicates that the number is reproduced by us since the number is not provided in the original paper. Others are cited from the original paper.

| Method | OH | VLCS | PACS | DN | TI | Avg. |
|---|---|---|---|---|---|---|
| Zero-Shot | 82.4 | 81.7 | 96.1 | 56.6 | 33.8 | 70.1 |
| +*ours* | 87.0 | 85.1 | 98.1 | 61.1 | 59.7 | 78.2 *(+8.1%)* |
| CoOp[†] (Zhou et al., 2022b) | 83.0 | 80.8 | 96.4 | 59.5 | 46.8 | 73.6 |
| +*ours* | 87.3 | 85.3 | 98.2 | 61.0 | 59.9 | 78.3 *(+4.7%)* |
| CoCoOp[†] (Zhou et al., 2022a) | 83.4 | 80.3 | 96.7 | 59.4 | 45.3 | 73.2 |
| +*ours* | 87.4 | 84.8 | 98.4 | 61.3 | 58.8 | 78.2 *(+5.0%)* |
| DPL (Zhang et al., 2023a) | 84.2 | 84.3 | 97.3 | 56.7 | 52.6[†] | 75.0 |
| +*ours* | 87.2 | 85.1 | 98.0 | 61.4 | 60.6 | 78.5 *(+3.5%)* |
| DUPRG (Niu et al., 2022) | 83.6 | 83.9 | 97.1 | 59.6 | 42.0 | 73.2 |
| +*ours* | 87.0 | 85.4 | 98.2 | 61.5 | 60.1 | 78.4*(+5.2%)* |
| STYLIP[†] (Bose et al., 2024) | 84.1 | 84.8 | 96.8 | 59.9 | 57.4 | 76.6 |
| +*ours* | 87.5 | 85.3 | 98.5 | 62.1 | 59.9 | 78.7*(+2.1%)* |
| PromptStyler (Cho et al., 2023) | 83.6 | 82.9 | 97.2 | 59.4 | 44.2[†] | 73.5 |
| +*ours* | 87.7 | 86.1 | 98.3 | 62.0 | 60.6 | 79.0*(+5.5%)* |

## 5.3 COMPARISON WITH PREVIOUS STATE-OF-THE-ARTS

In Table 5, we compare our solution with previous state-of-the-art CLIP-based domain generalization methods. Besides, we also compare our method to baselines including zero-shot CLIP ("zero-shot"), linear probing of CLIP ("Linear-Probe"), and standard full fine-tuning of CLIP with all source domains ("ERM-FFT"). We report the accuracy for each dataset and the average accuracy across all the datasets in Table 5 for comparison. [†] indicates that the number is reproduced by us since it is not provided in the original paper. Other numbers are cited from the original paper. For our method, we report numbers obtained by attention head purification combined with prompt learning ("Ours").

From Table 5, we find that CLIP's zero-shot result serves as a strong baseline. Directly linear probing or full fine-tuning CLIP's image encoder even results in worse accuracy, demonstrating that it is non-trivial to harness CLIP for domain generalization tasks. For methods that fine-tune the image encoder, existing competitors, including GESTUR, MIRO, and CLIPood, mainly focus on avoiding knowledge forgetting of CLIP during task adaptation. Our method outperforms these methods, *e.g.,* outperforming MIRO by more than 5%, indicating that our way of performing attention head purification is more effective on improving the CLIP's domain generalization performance. Overall, compared with various CLIP-based generalization methods, our method achieves the best result.

## 5.4 VISUALIZATION

We visualize the overall attention maps (Chefer et al., 2021) before and after attention head purification in Figure 3(a). We observe that the zero-shot model owns a sparser attention and pays more attention to the background which may not be generalizable across domains. In contrast, ours attends to the most discriminative and generalizable properties of the object for classification. We further visualize the attention maps generated by specific attention heads. We show the attention maps of the top two heads (Figure 3(b)(c)) and the last head (Figure 3(d)) ranked by the learned head gates. The top two

Table 5: Comparison with previous state-of-the-arts. [†] indicates that the number is reproduced by us as it is not reported by the original paper. Others are cited from the original paper.

| Method | OH | VLCS | PACS | DN | TI | Avg. |
|---|---|---|---|---|---|---|
| Zero-Shot | 82.4 | 81.7 | 96.1 | 56.6 | 33.8 | 70.1 |
| Linear-Probe[†] | 79.3 | 77.5 | 94.9 | 48.2 | 44.6 | 68.9 |
| ERM-FFT[†] | 80.0 | 79.1 | 91.4 | 53.9 | 44.1 | 69.7 |
| CoOp[†] (Zhou et al., 2022b) | 83.0 | 80.8 | 96.4 | 59.5 | 46.8 | 73.6 |
| CoCoOp[†] (Zhou et al., 2022a) | 83.4 | 80.3 | 96.7 | 59.4 | 45.3 | 73.2 |
| MaPLe[†] (Khattak et al., 2023) | 83.4 | 82.2 | 96.5 | 59.5 | 50.2 | 74.4 |
| VPT[†] (Jia et al., 2022) | 83.2 | 82.0 | 96.9 | 58.5 | 46.7 | 73.6 |
| DPL (Zhang et al., 2023a) | 84.2 | 84.3 | 97.3 | 56.7 | 52.6[†] | 75.0 |
| DUPRG (Niu et al., 2022) | 83.6 | 83.9 | 97.1 | 59.6 | 42.0 | 73.2 |
| PromptStyler (Cho et al., 2023) | 83.6 | 82.9 | 97.2 | 59.4 | 44.2[†] | 73.5 |
| STYLIP (Bose et al., 2024) | 84.6 | **86.9** | 98.1 | 62.0 | 57.4[†] | 77.8 |
| DSPL (Cheng et al., 2024) | 86.1 | 86.4 | 97.5 | 62.1 | 57.1 | 77.8 |
| SPG (Bai et al., 2024) | 83.6 | 82.4 | 97.0 | 60.1 | 50.2 | 74.7 |
| VL2V-SD (Addepalli et al., 2024) | 85.4 | 82.7 | 95.7 | 58.7 | 41.2 | 72.7 |
| CLIP-LoRA[†] (Zanella & Ben Ayed, 2024) | 83.9 | 83.1 | 97.1 | 58.4 | 55.7 | 75.6 |
| GESTUR (Lew et al., 2023) | 84.2 | 82.8 | 96.0 | 58.9 | 55.7 | 75.5 |
| MIRO (Cha et al., 2022) | 82.5 | 82.2 | 95.6 | 54.0 | 54.3 | 73.7 |
| CLIPood (Shu et al., 2023) | 87.0 | 85.0 | 97.3 | **63.5** | 60.4 | 78.6 |
| CLIPood[†] (Shu et al., 2023) | 85.3 | 83.6 | 97.3 | 59.5 | 58.7 | 76.9 |
| Ours | **87.7**$_{\pm0.1}$ | 86.1$_{\pm0.3}$ | **98.3**$_{\pm0.2}$ | 62.0$_{\pm0.1}$ | **60.6**$_{\pm0.4}$ | **79.0** |

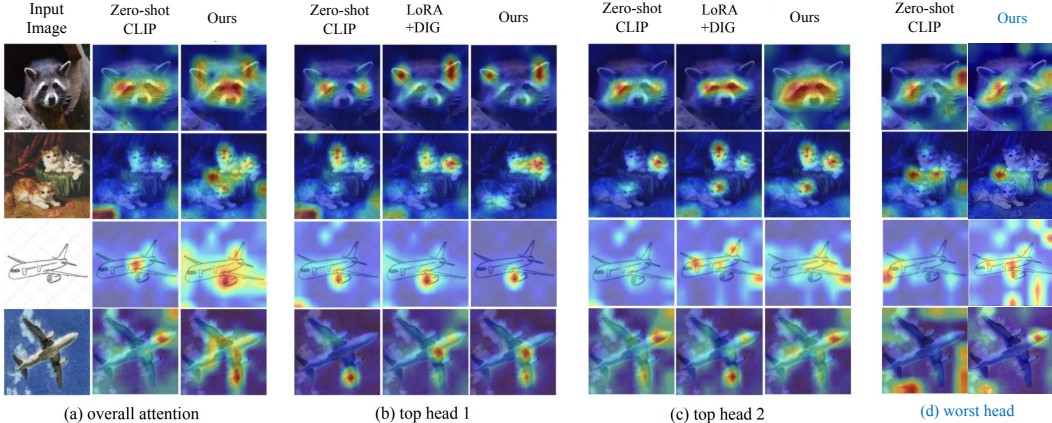

(a) overall attention      (b) top head 1      (c) top head 2      (d) worst head

Figure 3: Attention maps for target samples in DomainNet.

heads exhibit different preferences. Generally, top head 1 tends to capture the most discriminative part of each object, for example, the ear of a raccoon (first row) and the engine of a plane (third row), while top head 2 tends to focus on the overall region of an object. Besides, compared with purifying heads using conventional LoRA with DIG (middle column in Figure 3 (b)(c)), thanks to the interference elimination of HA-LoRA, our method can better highlight the perceptual tendency of a specific head. For the worst head, as shown in Figure 3(d), background areas receive high attention. By assigning less weight to such heads in our method, their influence is suppressed.

# 6 CONCLUSION

In this paper, we propose a simple yet effective method to harness CLIP for domain generalization, *i.e.,* attention head purification. Specifically, we perform attention head purification from two perspectives including task-level purification and domain-level purification. For task-level purification, we design head-aware LoRA to make each head specifically adapted to the downstream tasks. For domain-level purification, we adopt a domain-invariant gating strategy to encourage the model to select/emphasize the most generalizable attention heads. During training, we jointly perform the task-level purification and the domain-level purification. Experiments on five representative domain generalization benchmarks demonstrate the superiority of our proposed method.

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

## A  APPENDIX

### A.1  ATTENTION HEAD EVALUATION STRATEGY MENTIONED IN FIGURE 1

In Figure 1(a), we evaluate the importance of attention heads for domain generalization tasks with different strategies including "randomly drop", "manually drop", "drop by cross-validation", and "adapt & drop". We introduce the implementation details of each strategy as follows:

**Randomly drop**: we randomly sample a certain number of attention heads and drop them for inference. We repeat the random sampling three times and report the average results.

**Manually drop**: we manually evaluate the generalization performance of each head based on the text describing the properties of each head provided in Gandelsman et al. (2023) and then drop the least generalizable heads for inference.

**Drop by cross-validation**: Each time we select one domain out of available domains as the target domain for evaluation. We use the remaining domains to train the model to learn the importance of each head for domain generalization task. Specifically, we adopt $\mathcal{L}_{cls}$ to train a learnable Bernoulli gate for each head representing the probability to remain the head, with the help of Gumbel-Softmax trick (Huijben et al., 2022). During inference, we drop heads with probability from lowest to highest. The CLIP is frozen during training. We report the average accuracy over different target domains.

**Adapt & drop**: following the above cross-validation setup, we perform attention head adaptation with LoRA and learn the gates simultaneously.

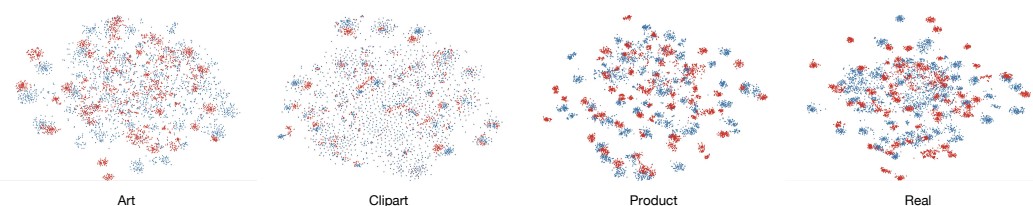

Art    Clipart    Product    Real

Figure 4: t-SNE visualization of features from target domain on OfficeHome. We compare the visualization results of the original feature (blue dot) and our method (red dot).

Table 6: **Left:** Effect of soft gating in domain-level purification. **Right:** Training and inference time on DomainNet. Experiments were conducted on a single RTX4090 24Gb with the original code provided by the authors.

| Gating Strategy | PACS | OfficeHome | DomainNet |
|---|---|---|---|
| binary mask | 96.9 | 85.4 | 57.7 |
| soft gating | 98.1 | 87.0 | 61.1 |

| Method | Training time | Inference time |
|---|---|---|
| CoOp | 2h 57min. | 39s |
| DPL | 2h | 41s |
| MIRO | 5h 24min. | 39s |
| CLIPood | 4h 46min. | 39s |
| Ours | 1h 30min. | 39s |

## A.2 Performing DIG using binary mask instead of soft gating

With the help of Gumbel-Softmax trick, we can generate binary mask for attention heads to fully retain or remove a specific head. In Table 6 (Left), we provide the results of replacing soft gating with binary mask in DIG. We find that soft gating yields better performance.

## A.3 Visualization with t-SNE

We present the t-SNE visualization of the feature distribution on OfficeHome in Figure 4. The blue dots denote CLIP's original visual feature and the red dots denote visual features generated by our method. For the original visual features, feature distribution is more dispersed especially for the domain Clipart. Nevertheless, benefiting from the attention head purification, the features of ours are more compact and the distribution is more concentrated, which is in line with the superior domain generalization performance of our method.

## A.4 Computational efficiency

Table 6 (Right) compares the training time of the leading prompt-learning methods (CoOp (Zhou et al., 2022b) and DPL (Zhang et al., 2023a)) and fine-tuning methods (MIRO (Cha et al., 2022) and CLIPood (Shu et al., 2023)). Our method achieves better performance with shorter training time. We evaluate the inference time for each method on the "Real" domain of DomainNet dataset, with batch size set to 128. We observe that our method doesn't introduce additional inference time compared to previous works.

## A.5 Effect of Fine-tuning the text encoder

The text features with specifically designed prompt in nature contain rare domain-specific information. Additionally, co-adapting the image encoder and text encoder on limited data can lead to overfitting, potentially disrupting the alignment between image and text features. As a result, we freeze the text encoder. As shown in Table 7, co-adapting the image and text encoders results in notable performance degradation.

Table 7: Effect of fine-tuning CLIP's text encoder. We compare the results of co-adapting image encoder and text encoder with those of ours which freezes the text encoder.

| Method | OH | PACS |
|---|---|---|
| Ours w. fine-tuning text encoder | 84.7 | 96.5 |
| Ours w.o. fine-tuning text encoder | 87.0 | 98.1 |

### A.6 VISUALIZATIONS WITH OVERALL ATTENTION

In Figure 5, we compare the overall attention maps of Head-Aware LoRA with DIG (our method), conventional LoRA with DIG, and the regularization-based method CLIPood. Our method generates better attention, enabling a more accurate and comprehensive perception of discriminative parts of the target object.

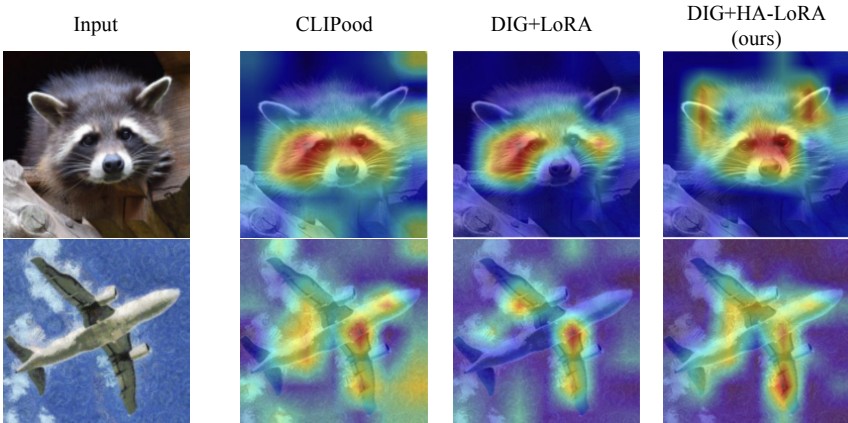

Figure 5: Comparison of the overall attention. Images are sampled from the DomainNet dataset. The "Real" is selected as the target domain while others are selected as source domains to train the CLIP.

### A.7 VISUALIZATION WITH LEARNED GATING WEIGHTS OF DIG

In Figure 6, we visualize the learned gating weights of DIG. The observation is that the weight distribution is non-uniform, and there exists an apparent gap between the largest weight and the smallest weight. The results demonstrate that with DIG, some heads are relatively emphasized and some heads are relatively suppressed.

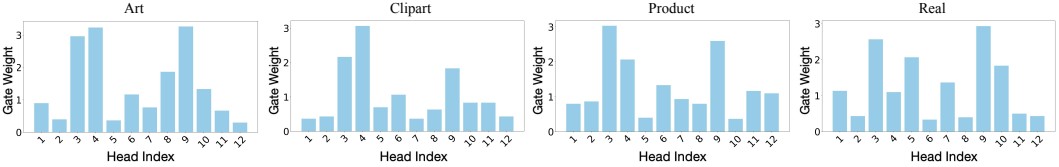

Figure 6: Distribution of the learned gating weights of DIG. Models are trained on OfficeHome. The domain name refers to the target domain in each case. The gate weights are not within 0-1 since we multiply the gates after Softmax operation $\hat{g}$ by the number of heads $\gamma$, as shown in Equation (5).

### A.8 EFFECTIVENESS OF DIG IN REMOVING DOMAIN-SPECIFIC INFORMATION.

In Figure 7, we provide attention to show the effectiveness of DIG in removing domain-specific information. The attention of different heads is ranked by the weights of DIG. For illustration purposes, we visualize the overall attention by aggregating the attention from all the heads. Purely

using HA-LoRA, the weights of different head attentions are equal to 1 for aggregating, while with DIG, the weights of different heads are different. From DIG ranking, we observe that the attention map which focuses on the discriminative parts of target object owns a larger gate weight while the attention map which focuses on the object-irrelevant background area owns a smaller gate weight. As a result, we observe that with the application of DIG, the domain-specific components (*e.g.*, background regions) in the overall attention are effectively suppressed, allowing better focus on the target object (*e.g.*, the monkey).

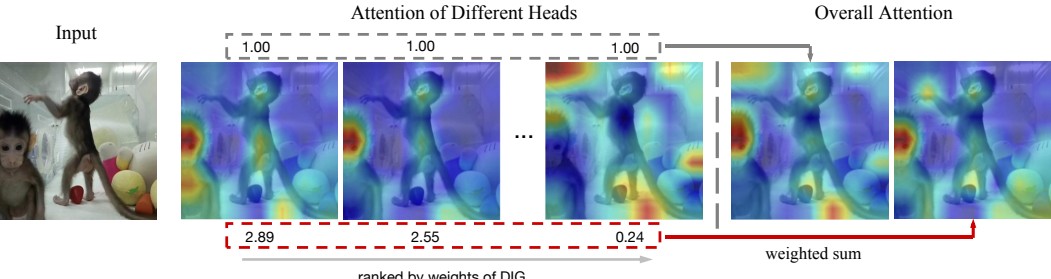

Figure 7: Effectiveness of DIG in removing domain-specific information. For illustration purpose, the overall attention is computed as a weighted sum of attention from different heads, using either DIG weights (as shown in red color) or uniform weights (*i.e.*, without DIG, as shown in gray color).