# OpenReview forum: "Attention Head Purification: A New Perspective to Harness CLIP for Domain Generalization"
_ICLR.cc/2025/Conference — Submitted to ICLR 2025_

### Official Review · Reviewer_L69A · 2024-10-24

**Soundness:** 3
**Presentation:** 3
**Contribution:** 2
**Rating:** 5
**Confidence:** 5

**Summary:**

This paper addresses the problem of domain generalization (DG), where data from multiple source domains is used during training, with the objective of evaluating the trained model on unseen target domains. The study focuses on CLIP ViT models, drawing on previous findings that different attention heads capture distinct properties of an image. The authors introduce a head-aware LoRA mechanism, specifically tailored for each attention head, to capture task-specific knowledge while minimizing interference. Additionally, a domain-invariant gating module with learnable gating values is applied to the head outputs. These outputs are weighted, and a Maximum Mean Discrepancy (MMD) loss is employed to promote the extraction of domain-invariant features. The proposed approach is validated on zero-shot CLIP models and can be seamlessly integrated with other DG methods.

**Strengths:**

1.	The strategy of reducing interference among attention heads and decoupling the learning of task-specific and domain-invariant features is technically sound.
2.	The proposed method is efficient, as it introduces only a small number of learnable parameters.
3.	The method demonstrates versatility by being compatible with various approaches, and notable improvements have been observed through its integration.

**Weaknesses:**

1.	The motivation of the paper is based on the observation that different attention heads capture distinct properties. However, the CLIP model, while trained as a foundation model, is not flawless. Its generalized knowledge may not cover specific datasets comprehensively, limiting its ability to attend perfectly to every domain. The proposed method offers a more parameter-efficient fine-tuning approach by applying LoRA independently to each attention head. The observed improvements can be attributed to the model's ability to capture unique properties from the source domains, which in turn enhances performance on the target domain. Therefore, the claim of "purifying at the task level" may not be entirely appropriate, as the HA-LoRA mechanism complements the existing knowledge within the CLIP model rather than isolating it.
2.	Why does the proposed method modulate only the projection matrices of Q and V with HA-LoRA? Since the attention map is generated through the interaction between Q and K to query the information stored in V, wouldn’t it be more consistent to also modulate the K projection? Clarifying the rationale behind this design choice would strengthen the paper.
3.	The paper does not provide an ablation study on the impact of sharing the B matrix, which would be valuable for understanding its role in the proposed approach.
4.	The domain-invariant gating mechanism appears to function as an additional layer of attention applied on top of the attention heads. It is unclear how this linear scaling effectively filters out domain-specific information while preserving only the domain-invariant features. A more detailed explanation of this mechanism would help clarify its effectiveness.
5.	In Table 4, all the compared and integrated methods use a frozen image encoder. It would be insightful to include results for these methods with a naive LoRA applied, ensuring a fair comparison and better understanding of the proposed approach's advantages.

**Questions:**

1.	Since the best performance of the proposed method is achieved when integrated with PromptStyler, how exactly is PromptStyler incorporated in Table 4? Given that PromptStyler operates under a source-free DG setting and does not require access to source data, it may not be appropriate to directly reuse results from the original paper without further justification.
2.	How is "importance" defined when evaluating the attention quality, as shown on the right side of Figures 1 and 3? Additionally, it would be insightful to visualize how the proposed method modulates the attention by comparing attention maps before and after applying the method for the same head.
3.	What does the "worst" attention look like for the proposed method, as indicated in Figure 3? Providing such an example would offer a deeper understanding of the method's limitations.
4.	How does the proposed method compare in terms of the number of learnable parameters? A direct comparison would highlight the parameter efficiency of the approach.
5.	What do the learned gating weights look like? Visualizing or analyzing these weights could provide more insight into the gating mechanism's behavior.
6.	How sensitive is the method to the number of layers where HA-LoRA is integrated and to the rank numbers, as mentioned in Lines 354-355? An ablation study on these aspects would help clarify the impact of these choices.
7.	Can the authors further elaborate on the statement: "CLIP, by design, may contain domain-specific cues that constrain its domain generalization performance"? A more detailed interpretation would strengthen the argument.
8.	If HA-LoRA primarily focuses on task-level adjustments, it may still retain biased domain knowledge, as domain-specific information is only removed at a later stage. Is there a concrete example that demonstrates this process and illustrates how the method addresses or mitigates this bias?

---

### Official Review · Reviewer_wUHR · 2024-11-02

**Soundness:** 2
**Presentation:** 3
**Contribution:** 2
**Rating:** 5
**Confidence:** 4

**Summary:**

This paper focuses on domain generalization of CLIP. It introduces task-level and domain-level purification to make attention heads more task-adapted and domain-generalizable.

**Strengths:**

The proposed method is simple and clearly presented.

**Weaknesses:**

1. Although the method is technically simple, its final performance is not as promising as expected, as shown in the SOTA table (Table 5).
2. The paper distinguishes the proposed method from previous ones by posing the question: "Is the way to avoid knowledge forgetting sufficient to harness CLIP for domain generalization?" Does this imply that the proposed method can complement previous knowledge-forgetting avoidance techniques, such as regularization-based methods? If so, it would be beneficial to show this experimentally.
3. The visualization experiment in Figure 3 does not fully verify the motivation. To demonstrate the attention head purification resulting from the proposed design, it would be more convincing to compare head-aware LoRA with conventional LoRA, as well as the proposed method with regularization-based methods.

**Questions:**

Please refer to Weakness 2.

---

### Official Review · Reviewer_N2Qg · 2024-11-03

**Soundness:** 3
**Presentation:** 3
**Contribution:** 2
**Rating:** 3
**Confidence:** 3

**Summary:**

The paper presents an approach to harness CLIP for DG, highlighting that some attention heads may encode domain-specific information that limits generalization capabilities. The authors introduce two levels of purification: task-level and domain-level. For task-level purification, they propose head-aware LoRA (HA-LoRA) to adapt each attention head to the specific task, while domain-level purification involves a learnable gating strategy to select heads that contribute to domain-invariant features.

**Strengths:**

1. The paper proposes a novel CLIP-based method with the attention head purification, which provides new perspectives for research on domain generalization.
2. Extensive experiments demonstrate the effectiveness of the method.
3. The paper is well organized and easy to follow.

**Weaknesses:**

1. Limited exploration of head interactions: The focus on purifying heads independently may overlook potential interactions between heads, as attention heads often work in conjunction. Neglecting their interactions may lead to suboptimal configurations and reduce the overall efficacy of the model.
2. Limited theoretical justification: The paper lacks a theoretical underpinning for why attention head purification, specifically through HA-LoRA, would systematically improve DG. The concept of different attention heads being more or less generalizable is intuitive but could benefit from further theoretical exploration or a more rigorous explanation of why the proposed method is expected to work universally across domains and tasks.
3. Gating complexity: The gating mechanism, while useful, introduces additional complexity that may complicate the model's interpretability and practical implementation. This paper does not provide a detailed analysis of the computational cost of the proposed method compared to other methods, which could be important for its practical applicability.
4. Unexplored interaction with text encoder: Since CLIP’s strength lies in its vision-language alignment, the text encoder is kept fixed throughout the training, which seems to avoid investigating potential improvements that could arise from co-adapting the text and image encoders for DG tasks.

**Questions:**

1. Analysis of computational overhead: Please provide more details on the computational costs of your method. A comparison of runtime and memory usage between your method and other CLIP-based approaches would be helpful.
2. Additional explanation of gating strategy: It would be beneficial to provide more details regarding the impact of the gating strategy, potentially including ablation studies. As it stands, this does not appear to be a particularly novel idea.
3. Experimental results on MMD: Table 2 shows a significant drop in performance when MMD loss is integrated with HA-LoRA. The HA-LoRA module is designed to help the model retain task-relevant knowledge, and the MMD loss aims to capture information that is consistent across multiple domains. These two objectives should not be inherently conflicting; however, their combination leads to a decline in performance.
4. Further theoretical insights: Could you provide more theoretical insights into why head-specific purification specifically improves DG compared to other forms of fine-tuning?

---

### Official Review · Reviewer_2Up5 · 2024-11-03

**Soundness:** 3
**Presentation:** 3
**Contribution:** 3
**Rating:** 5
**Confidence:** 4

**Summary:**

This paper addresses the challenge of Domain Generalization by observing that domain-specific cues inherent in CLIP can hinder generalization. To tackle this, the authors introduce a novel strategy  including two levels of purification of attention head: task-level, achieved through head-aware LoRA to make heads more task-adapted, and domain-level, using a gating strategy with MMD loss to make selected heads more domain-invariant. Extensive experiments on several datasets show that this method outperforms previous state-of-the-art methods, validating its effectiveness.

**Strengths:**

1. The paper is well written and easy to follow.

2. The authors provide a valuable observation that not all attention heads within CLIP contribute equally to domain generalization in specific tasks.

3.  The paper presents a structured approach by decoupling attention head purification into two levels: task-level and domain-level purification.

4.  Extensive experiments on well-recognized domain generalization benchmarks provide robust evidence of the method's effectiveness.

**Weaknesses:**

1. Insufficient Explanation of HA-LoRA: The proposed HA-LoRA claims to improve task-level generalization by adding independent parameters for each head. However, adding these additional parameters typically risks overfitting. The authors should clarify why overfitting does not appear to be an issue in this case. Additionally, aside from the independent B matrices, no specific training objective is proposed to encourage task-level purification. It seems implausible that the independent B matrices alone could achieve this effect. Providing more insight into how independent LoRA B matrices promote task-specific learning would strengthen the argument.

2. Parameter Increase Disclosure: Although the authors mention additional parameters in LoRA, there’s no concrete information on the parameter increase relative to the original LoRA. This is crucial for understanding the model’s computational trade-offs.

3. Lack of reproduced results: In Table 4, the performance marked as 'xx method + ours' were obtained by the authors. To better understand the impact of HA-LoRA, please also report the baseline performance of these prompt-based methods as reproduced by the authors. This should not be difficult as the authors should first reproduce the results of original papers and then conduct further analysis by adding their proposed part.

4. Related Work: The author should also discuss with the paper regarding the proposed HA-LoRA: Tian, Chunlin, et al. "HydraLoRA: An Asymmetric LoRA Architecture for Efficient Fine-Tuning." arXiv preprint arXiv:2404.19245 (2024).

**Questions:**

1. Parameter Efficiency: Can the authors provide quantitative evidence on the added parameter count for HA-LoRA compared to original LoRA, and how this scales with model size and number of tasks?

2. Impact of Random Seeds: Given that slight variations in random seeds can impact model performance, how sensitive is this method to random seeds? Have the authors performed robustness tests across different seeds?

3. Empirical Evidence for Task-Specific Knowledge in B Matrices: What empirical evidence supports the claim that each independent B matrix learns task-specific information? For example, could visualizations or analysis of learned representations clarify how these matrices differ across tasks?

---

> ### Comment · Reviewer_2Up5 · 2024-11-27
>
> Thanks for the authors' feedback.
>
> Based on the first response above, it appears that task-level purification should not be considered separate from the DIG approach. In this context, I believe the paper only proposes a router/gating mechanism to select heads when generalizing to a specific domain. Despite the simplicity of the method, router/gating has been extensively studied in the literature, and there is no specific design of this component for the domain generalization problem. As such, I believe this paper is not yet ready for publication. The authors should conduct a more in-depth study to provide additional insights and refine their method accordingly. Lastly, I recommend removing the phrase "task-level purification," as it does not accurately reflect a process of literal "purification."
>
> Based on the above reason, I will maintain my current rating.

---

### Meta-Review · Area_Chair_wfh9 · 2024-12-17

**Metareview:**

This paper proposes a new perspective (attention head purification) to harness CLIP for domain generalization (DG). The paper finally got four negative scores.

The strengths of this paper include: 1) well-written; 2) provided a valuable observation; 3) a structured approach; 4) the method is effective on DG benchmarks.

However, the reviewers think this paper has the following drawbacks: 1) insufficient explanation of HA-LoRA and head interactions; 2) parameter increase disclosure; 3) lack of reproduced results; and 4) limited theoretical justification.

The authors have provided a rebuttal. After checking the rebuttal and comments, the reviewers acknowledged that some of their concerns have solved but the explanation of HA-LoRA is not satisfied. Finally, the reviewers think this paper did not well explain why the proposed method is specifically designed for the domain generalization problem and consistently gave negative scores. To this end, AC thinks this paper cannot meet the requirement of ICLR at this point and thus regrets to recommend rejection.

**Additional Comments On Reviewer Discussion:**

The authors have provided a rebuttal. After checking the rebuttal and comments, the reviewers acknowledged that some of their concerns have solved but the explanation of HA-LoRA is not satisfied. Finally, the reviewers think this paper did not well explain why the proposed method is specifically designed for the domain generalization problem and consistently gave negative scores.

---

### Decision · Program_Chairs · 2025-01-22

Reject